# Risk Assessment of Pesticide Residues by GC-MSMS and UPLC-MSMS in Edible Vegetables

**DOI:** 10.3390/molecules28031343

**Published:** 2023-01-31

**Authors:** Mohamed T. Selim, Mohammad M. Almutari, Hassan I. Shehab, Mohamed H. EL-Saeid

**Affiliations:** 1Institute of Research and Consultancy, Ministry of Environment, Water and Agriculture, Riyadh 11442, Saudi Arabia; 2National Agriculture & Animal Resources Research Center, Riyadh 11442, Saudi Arabia; 3Plant Resource, Ministry of Environment, Water and Agriculture, Riyadh 11442, Saudi Arabia; 4Food and Environmental Research Center in Unayzah, Ministry of Municipal and Rural Affairs, Unayzah 51911, Al-Qassim, Saudi Arabia; 5Chromatographic Analysis Unit, Soil Science Department, College of Food & Agricultural Sciences, King Saud University, P.O. Box 2460, Riyadh 11451, Saudi Arabia

**Keywords:** pesticide residues, risk assessment, QuEChERS, EDIs, ADI, HRI, GC–MS/MS, LC-MSMS

## Abstract

In recent years, there has been a significant increase related to pesticide residues in foods, which may increase the risks to the consumer of these foods with the different quality and concentrations of pesticide residues. Pesticides are used for controlling pests that reduce yields. On the other hand, it has become a major public health concern due to its toxic properties. Thus, the objective of the current study employed the application of Quick Easy Cheap Effective Rugged Safe (QuEChERS) method, in combination with gas and liquid chromatography-tandem mass spectrometric detection (GCMSMS, LCMSMS) in order to determine 137 pesticide residues (63 insecticides, 41 acaricides, 40 herbicide, 55 fungicide, nematicide, growth regulator, Chitin synthesis inhibitors, and Juvenile hormone mimics), in 801 vegetables such as 139 tomatoes, 185 peppers, 217 squash, 94 eggplants, and 166 cucumbers from different locations in Hail and Riyadh cities. The results showed that the majority of pesticide residues were detected for each of the following pesticides: acetaimpride, metalaxyl, imidaclopride, bifenthrin, pyridaben, difenoconazole, and azoxystrobien, which were repeated in the samples studied 39, 21, 11, 10, 8, 7, and 5, respectively. In addition, results observed that the tomato was the most contaminated with pesticide residues; it was contaminated with 19 compounds and was followed by pepper, cucumber, and squash, and the last commodity in the contaminated ranking was eggplant. The highest calculated estimated daily intakes (EDIs) were recorded for tomatoes which were estimated between 0.013 to 0.516 mg/kg of body weight per day (bw/day) while the lowest EDIs value was between 0.000002 to 0.0005 mg/kg of bw/day for cucumber. Results indicated that the EDIs values were lower than the acceptable daily intake (ADI) values. Results observed that the most of pesticide residues exposure in food consumption in Saudi Arabia were lower than ADIs. In addition, the highest value for health risk index (HRI) was recorded with Ethion residue in tomato, but in sweet pepper, the highest value for HRI was 127.5 in the form of fipronil residue. On the other hand, results found that the highest values of HRI were 1.54, 1.61, and 0.047 for difenoconazole, bifenthrin, and pyridaben residues in squash, eggplant, and cucumber.

## 1. Introduction

In recent years, we have observed a substantial increase in the importance placed on aspects related to pesticide residues and a growing demand for better agricultural practices, transparency, and traceability in the production and marketing of conventional food. On one hand, pesticides make it possible to increase food production by destroying weeds and pests that attack cultivable plants and agricultural crops, and they also limit losses sustained during the transport and storage of food. On the other hand, pesticides are one of the most dangerous chemical compounds due to their toxic properties, environmental persistence, and bioaccumulation capability. Thus, the presence of pesticide residues in food commodities is a source of great worry; what makes it more complex is that some of these vegetables are consumed fresh or semi-processed, which may contain elevated levels of chemicals compared to other food crops of plant origin. Exposure to pesticides through diet is thought to be five orders of magnitude higher than other exposure routes, for example, air and drinking water [1,2,3]. The level of pesticide residues in foodstuffs is generally legislated so as to minimize consumers’ exposure to harmful or unnecessary pesticide intakes, their maximum concentrations are controlled by the European Union Council Directive 91/414/EEC [4], and established maximum residue limits (MRLs) for pesticides in foodstuffs and animal feed in Directive No. 396/2005 (Regulation2005) [5].

LC and GC coupled to MS/MS detection provides accurate methods of identifying and quantifying numerous pesticides in food extracts. Several articles have recently been published where these techniques were successfully utilized for the analysis of pesticides in fruits and vegetables. Due to the high selectivity provided by MS/MS detection, simple extraction techniques with little cleanup are employed [6]. 

In the last few years, the so-called QuEChERS (Quick, easy, cheap, effective, rugged, and safe) sample preparation procedure has become a widely used technique because of its applicability on a wide range of pesticides [7,8,9,10]. It has several advantages over traditional methods of pesticide residue analysis, for example, high recoveries (>85%) are achieved for a wide polarity, very accurate (true and precise) results are achieved, solvent usage and waste are very small, and the MeCN is added by dispenser to an unbreakable vessel that is immediately sealed, thus, solvent exposure to the worker is minimal, the method is very inexpensive [11,12,13,14,15]. 

In this study, we aimed to apply the QuEChERS methodology in combination with gas and liquid chromatography-tandem mass spectrometric detection (GC-MS/MS, LC-MS/MS), for the analysis of 137 pesticides, to determine residues of chemical pesticides (Organophosphates, OPs; acaricides, ACs; fungicides, FUs and insecticides of biological origin, INsB) used in vegetable farming in Hail and Riyadh cities; and to assess the health risk of adults due to the ingestion of pesticides in and on their vegetables.

## 2. Results and Discussion

### 2.1. Pesticide Residues in Raw Foods

A wide range of pesticide residues (63 insecticides, 41 acaricides, 40 herbicide, 55 fungicide, nematicides growth regulators Chitin synthesis inhibitors and Juvenile hormone mimics) in 801 vegetables such as 139 tomatoes, 185 peppers, 217 squash, 94 eggplants, and 166 cucumbers from different locations in Hail and Riyadh cities were detected in the Kingdom of Saudi Arabia during 2020. Regarding the pesticides that were screened, results showed that most of the pesticide groups that were detected belonged to different groups as fungicides (10 compounds), insecticides (8 compounds), acaricides (2 compounds), and multifunctional groups, such as insecticides/acaricides (4 compounds), insecticides/IGR (1 compound), and insecticides/nematicides (one compound). These compounds belong to many chemicals groups, as we found that the most frequent chemical group was the Triazole chemical group which has three compounds (penconazole, propiconazole, and triadmenol) as a fungicide with a percentage to reach 38%. Following this, each of the other groups (carbamate, dicarboximide, neonicotinoid, organophosphate and pyrethroid) were repeated twice. On other hand, the other remaining groups (phenylpyrazoles, chlorophenyl, dioxolanes, Hydrazine carboxylate, hydroxyanilides, methoxyacrylates, oxadiazine, phenylamide, pyridazinone, quinazoline, tetronic acid, triazolinthione, and unclassified) were repeated one time. All of these chemical groups use 31% insecticides. Following in the most frequency is the mixed group of insecticides/acaricides with 15%. After that, there is the acaricides group with 8%, and the last groups both insecticides/acaricides and fungicides/nematicides have 4% for both. Figure 1. Data was mentioned previously partially in agreement with [1,2,3]. 

In our study, we observed the represented data in Table 1 and Figure 1, Figure 2 and Figure 3 and the majority of residue compound was detected to be acetaimpride, followed with metalaxyl, imidaclopride, bifenthrin, pyridaben, difenoconazole, and azoxystrobien with a frequency of 39, 21, 11, 10, 8, 7, and 5, respectively. After that, triadmenol, ethion, deltamethrin, tolclofos-meth, spiromesifen, propiconazole, penconazole, fenitrothion, bifenazate, and buprofezine had frequencies of 4, 3, 3, 3, 2, 2, 2, 2, 2, 2, and 2. Finally, tebuconazole, procymidone, oxamyl, methomyl, indoxacarb, fipronil, fenhexamid, and fenazaquin had frequencies of 1 for all previous compounds, respectively.

On the other hand, we observed that the commodity in our study most contaminated with pesticide residues was tomato, as it is contaminated with acetaimpride, azoxystrobien, bifenthrin, bifenazate, deltamethrin, difenoconazole, ethion, fenazaquin, fenhexamid, fenitrothion, imidaclopride, indoxacarb, iprodione, metalaxyl, oxamyl, propiconazole, pyridaben, tolclofos-meth, and triadmenol with a frequency of 19 times followed by the frequency of 16 times for pepper, which was contaminated with acetaimpride, azoxystrobien, buprofezine, bifenthrin, bifenazate, deltamethrin, difenoconazole, ethion, fipronil, imidaclopride, iprodione, metalaxyl, pyridaben, spiromesifen, tolclofos-meth and tebuconazole after that in ranking cucumber was contaminated with acetaimpride, azoxystrobien, difenoconazole, metalaxyl, methomyl, penconazole, procymidone, and pyridaben. Following with a frequency of 8 times, squash was contaminated with acetaimpride, azoxystrobien, bifenthrin, metalaxyl and penconazole. With a frequency of 6 times, the last commodity in contaminated rankings was eggplant with a frequency of 4 times for acetaimpride, buprofezine, bifenthrin, and metalaxyl. Table 1, Figure 2 and Figure 3. Overall, the pesticide residues which were found in this study were approximately similar to other studies [8,9,11]. 

### 2.2. Estimation of Dietary Intake 

The objective of risk assessment from the point of view of food safety is, to ensure that in order to evaluate a dietary risk assessment, the ADI values were determined by summing the quotes of the pesticide ingested from various alimentary sources (i.e., vegetables and fruits). The Codex Alimentarius Commission of the FAO of the United Nations and WHO (FAO/WHO 2004) (17) recommended abiding with MRLs in fruits and vegetables. Monitoring of pesticide residues is a key tool for ensuring conformity with regulations and providing a check on compliance with good agricultural practice. The consideration of possible exposure to pesticide residues is an integral part of the risk assessment process to ensure that the ADI of the pesticides are not exceeded. As long as the residue of the pesticides ingested by consumers does not exceed the corresponding ADI, consumers are considered to be adequately protected. This is useful for assessing human exposure to pesticides through the food supply and for understanding the magnitude of health risks.

Additionally, the annual disappearance figures for a food commodity can be divided by the national population and by 365 days to obtain a “per capita” estimate of the food that is available for consumption per day expressed as grams per person per day (g/p/d). Disappearance data cannot be used to estimate intake for targeted sub-populations (e.g., young children, diabetics, or specific age-sex groups). The levels of contaminant pesticide residues used to estimate dietary intake of those substances can be obtained by combining the analytical results with amounts of food consumed reported in national food consumption surveys (Table 2). 

### 2.3. Estimation of Pesticide Exposure 

The estimated daily intake for each monitoring pesticide residue was calculated with the next formula: EDI = (commodity consumption × pesticide residue concentration)/body weight

### 2.4. Estimation of Health Risks from Pesticides

Estimation of the exposure risk to an adult person based on potential health risk by using the following formula: HRI = EDI/ADI

In our study, the authors compiled the available data on pesticide residues in different plants that generate food commodities, such as vegetables. On the basis of previously conducted studies in different cities of Saudi Arabia, it was possible to conduct a human risk assessment using the hazard risk index (HRI). The results are summarized in Table 2, for HRI assessment, the estimated daily intake (EDI) (mg/kg/day) and acceptable daily intake (ADI) values (mg/kg/day) were taken and calculated by following international guidelines [16,17,18,19,20], where EDI is the estimated average daily intake (mg/kg/day), C is pesticide residue concentration (mg/kg) multiplying by the food consumed, and W is the average weight of an adult. Reference values for the food consumption rate of vegetables and fruits were taken from literature as 0.3 kg/person/day for vegetables and 0.4 kg/person/day of fruits, respectively, while 60 kg was considered an average adult weight [21,22,23,24,25]. The HRI value for the risk estimation of different toxic metals and pesticides via food consumption was calculated, and the general consumption rates were used (regardless of seasonal and generic wise consumption) due to data scarcity (Figure 4, Figure 5, Figure 6, Figure 7 and Figure 8).

As we observed in Table 3, the calculated EDIs of tomatoes had been estimated between 0.013 to 0.516 mg/kg of bw/day. For sweet pepper, the EDIs value was between 0.0028 to 0.025 mg/kg of bw/day. However, in squash, the EDIs value was between 0.004 to 0.015 mg/kg of bw/day and in eggplant, the EDIs value was between 0.001 to 0.086 mg/kg of bw/day. Lastly, the EDIs value was between 0.000002 to 0.0005 mg/kg of bw/day in the cucumber. We observed that the EDIs values were lower than the ADI values. We reported that most pesticide residue exposure was lower than ADIs, and this depends on style of food consumption in Saudi Araba (Figure 4, Figure 5, Figure 6, Figure 7 and Figure 8). 

Furthermore, the EDIs values were used to estimate the hazard index (HRI) for each corps. We found a higher value for HRI for Ethion residue in tomato, but in sweet pepper, the higher value for HRI 127.5 was to fipronil residue. On the other hand, we found that the high value of HRI was 1.54 for difenoconazole residue in squash and 1.61 for bifenthrin residue in eggplant. Lastly, in cucumber, the high value was HRI 0.047 for pyridaben residue. We noticed all estimated data of the Hazard Index were exceeding the value of MRL, which may indicate a bad use of pesticides and failure to follow the application rates and the pre-harvest interval, which leads to exposure to health risks.

## 3. Materials and Methods

### 3.1. Chemicals and Reagents

Ultra-gradient HPLC-grade acetonitrile was purchased from J.T. Baker (Deventer, The Netherlands). Deionized water was obtained from a Milli-Q SP Reagent Water System (Millipore; Bedford, MA, USA). Formic acid (98% purity) and anhydrous magnesium sulfate were ordered from Fluka–Sigma–Aldrich (Steinheim, Germany). Each sample was filtered through a 13 mm × 0.45 um PTFE filter before injection, whilst PSA-bonded (primary secondary amine) silica was used as a sample clean-up step—both of them were from Supelco (Bellefonte, PA, USA). Acetic acid (Merck; Darmstadt, Germany) and sodium acetate-3-hydrate (Panreac; Castellonde Valles, Barcelona, Spain) were used for the sample preparation procedure. All certified pesticide standards obtained from Dr. Ehrenstorfer (Augsburg, Germany) were of 95 % or higher purity.

### 3.2. Study Area

This study was conducted in the Hail and Riyadh regions, which lie between longitude and latitude (43 N and 26 E and 34 N and 46 E), respectively. The city of Riyadh is characterized by a high population density, which is approximately six million people. On the contrary, the Hail region is characterized by a low population density, which reaches one and a half million people. These areas are dominated by a hot summer climate where temperatures reach 48 °C, and in winter, the average temperature drops to 9 °C.

### 3.3. Collection of Samples and Pretreatment

A total of 801 vegetable samples (139 tomatoes, 185 peppers, 217 squash, 94 eggplants, and 166 cucumber) from five local markets (three from Hail and two from Riyadh) were collected during the different seasons of 2019. Altogether, 2–3 units of fresh vegetables were collected from each local market (>1 kg) in accordance with the procedures described in the FAO, (1999). Samples were not rinsed. A portion of each sample, without tops such as the sepal and peduncle, was prepared according to annex I of European Commission regulation, 396/2005 EU (2010) using a knife and a chopping board and then thoroughly mixed. Two hundred gram of each sample were kept in a separate plastic bag at −20 °C until pesticide extraction and analysis could be carried out.

### 3.4. Extraction of Pesticide Residues by QuEChERS and Cleanup of Vegetables Samples

Vegetable samples were purchased from a local market and the preparation procedure was the same as the well-known and accepted QuEChERS (16), sample preparation procedure was applied to all the samples. After homogenization with the stainless-steel cutter (Sammic, Azpeitia, Spain), a 15 g portion of the homogenized sample was weighed in a 50 mL PTFE centrifuge tube. Then, 15 mL of acetonitrile were added with 6 g MgSO4and 2.5 g sodium acetate-3-hydrate and the samples were shaken vigorously by hand for 4 min. The extract was then centrifuged (3700 rpm) for 5 min. A 5 mL volume of the supernatant was removed to a 1-mL PTFE centrifuge tube containing 750 mg of MgSO4 and 250 mg of PSA. The extract was shaken in a vortex intensively for 20 s and centrifuged again (3700 rpm) for 5 min. Following this, an aliquot of the supernatant was evaporated under a nitrogen stream and reconstituted with acetonitrile/water (20/80) for LC analysis. Prior to injection into the LC–MS system, the sample was filtered through a 0.45-um PTFE filter. With this treatment, a 1 mL sample extract represents 1 g of sample.

### 3.5. Standard Preparation

A standard stock solution of each pesticide was prepared in acetonitrile at a concentration of 2000 µg/mL. A mixed standard solution was prepared at a concentration of 10 µg/mL from the individual stock solutions. The calibration curve for the LC measurements was prepared by diluting 10 µg/mL of the mixed standard solution to achieve final concentrations of 0.25, 0.5, 1, 2.5, 5, 10, 25, and 100 ng/mL in a mixture of acetonitrile and water (1:1, *v*/*v*). Stock and working solutions were stored at 4 °C until use. Pesticides were analyzed through Liquid Chromatography Triple Quadrupole Mass Spectrometry (LC-MSMS) and Gas Chromatography Triple Quadrupole Mass Spectrometry (GC-MSMS).

### 3.6. Analytical Techniques by Liquid Chromatography Triple Quadrupole Mass Spectrometry (LC-MSMS)

An ultra-high performance liquid chromatography (ACQUITY) coupled with a tandem quadrupole MS (XEVO TQD) was used with Mass Lynx 4.1 software (Waters Corporation, Milford, MA, USA). For the chromatographic separation, a reversed phase column, Atlantis T3 (100 × 3 mm, 5 µm), was used. The mobile phases (A and B) were water: methanol (98:2, *v*/*v*), and methanol, respectively, with 0.1% formic acid (FA) in each. The flow rate was maintained at 0.45 mL/min. The gradient program was initially set at 5% B (1 min), then linearly increased over the next 7.75 min to 100% and kept constant until 8.50 min. Thereafter, it was linearly decreased to 5%, and maintained for another 3.50 min (a total run time of 12 min). The MS was operated with Electrospray Ionization (ESI+). The optimized parameters included desolvation temperature (450 °C), desolvation gas flow rate (1000 L/hour), cone gas flow rate (50 L/hour), ion source temperature (120 °C), and capillary voltage (1 kV). The MS parameters are presented Table 4.

### 3.7. Compound Identification

Identification and confirmation of the target compounds on GC-MSMS, was performed by using the software (TraceFinder and Xcalibar) with an updated pesticides library consisting of a more than 900 pesticides and endocrine disruptors. The software incorporates the data such as retention time (with RT< ± 0.1 min), the parent/target ion (used for quantification), and 2 other ions (as qualifiers), for all the isomers, metabolites for almost all the included compounds (in the database).

MS analysis was carried out on a TSQ 8000 EVO GC triple stage quadrupole mass spectrometer. (Thermo Fisher Scientific, San Jose, CA). The MS conditions were as follows: Ionization mode: EI positive ion. Emission current: 50 μA. Ion source temperature: 220 °C. Scan type: SRM and Scan time: 0.02 s.

On the other hand, identification and confirmation of the target compounds on LC-MSMS, and two MRM transitions for each pesticide were generated using QUANPEDIA, ™. The data were acquired using MassLynx Software and processed using TargetLynx Application Manager. Peak shapes were adequate in most cases as shown in Figure 9.

### 3.8. Validation Design

The optimized analytical method was validated to ensure that it was fit for the intended purpose. The method was validated in terms of accuracy (mean recovery of the spiked samples at three different spiking levels), precision (intra-day and inter-day repeatability in terms of percent relative standard deviation, %RSD), selectivity, sensitivity (limit of quantitation (LOQ) and linearity (or linear range of measurement). The LOQ was calculated as the lowest concentration at which the recovery and precision was within the acceptable limits (recovery: 70–120%, precision: RSD < ±20%) (SANTE, 2019).

The calibration curve is determined by the analysis of each of the analysts at 6 calibration levels within the range of 0.25, 0.5, 1, 2.5, 5, 10, 25, and 100 ng/mL. The calibration curves were, in general, best fitted to a linear curve. The quantification was performed from the mean of three bracketing calibration curves. Most of the correlation coefficients (R2) were higher than or equal to 0.99.

For each level, three genuine replicates were performed. The method’s acceptance criteria were accuracy, precision, sensitivity, and the qualifier/quantifier ion ratio of the detected pesticides in real samples (to be <30%). The ion ratio was calculated as: ‘the mean ratio of the qualifier to quantifier ions for a pesticide calculated from an MMS batch’ subtracted from ‘the ion ratio for that pesticide in the positive sample’, and then dividing the resultant by the mean ion ratio calculated for the MMS of the same batch, the value thus obtained was multiplied by 100 to get the percentage value (SANTE, 2019).

To ensure the quality of the analytical work, the analytical batch was designed every time in a way to include a solvent/reagent blank, one matrix blank, and three replicates for all the three spiking levels. The solvent/reagent blanks were processed according to the complete extraction procedure under investigation, to eliminate any chances of laboratory and glassware contamination. One sample as matrix blank (extract of the sample viz. free of the targeted pesticides and which was used in the validation of the method) was also analyzed, and three replicates for all the three, i.e., highest spiking level (HSL), medium spiking level (MSL), and lowest spiking level (LSL), were also run in the same batch. The instrumental samples’ sequence was designed to be in the following order: reagent/solvent blank, then calibration standards in pure solvent, followed by matrix-matched standards (at the same concentration range as that of the standards-in-solvent) and then the real samples, bracketed by the standards-in-solvent, at the end. To eliminate the chances of carryover from previous samples’ injections, the instrument was also configured at back-flush settings, supported by an additional post-run column flushing of one minute.

Uncertainty (U) of the proposed multi-residue method was calculated by bottom-up empirical model in accordance with the ISO 21748. Uncertainty of the method’s repeatability, reproducibility, and trueness estimated was calculated as mentioned previously, partially in [5,10,13,15].

### 3.9. Pesticide Residue Analysis by Gas Chromatography Mass Spectrometry (GC–MSMS)

A Thermo Scientific TRACE 1310 Gas Chromatography coupled with TSQ 8000 Evo Triple Quadrupole Detector and AI 1310 Auto Sampler was used with Thermo Xcalibur 2.2 mass spectrometry data system (Software). For the chromatographic separation, a Thermo Scientific™ Trace GOLD™ TG-5SilMS 30 m × 0.25 mm I.D. × 0.25 µm film capillary column was used. The flow rate was maintained at constant flow 1.2 mL/min (He, inert carrier gas). The GC oven program was initially set at 70 °C (2 min), then increased 25 °C/min to 180 °C, 5 °C/min to 200 °C, and 10 °C/min to 280 °C, kept constant 5 min. The MS was operated with electrospray ionization (ESI+). The optimized parameters included transfer line (280 °C), electron energy(eV) 70, acquisition mode (SRM), and ion source temperature (320 °C). The Thermo ScientificTM TraceFinderTM software was used for method setup and data processing. For all pesticide compounds two SRM transitions were chosen for the overall MRM acquisition method. The first transition was used for quantitation, the second transition for confirmation. Table 5 and Figure 9, lists the SRM parameters for the compounds analyzed in this method.

## 4. Conclusions

High consumption of fruits and vegetables contaminated with pesticide residues above the MRL leads to a threat to the population’s health, and this is due to the poor handling practices for pests and disease control that also do not follow the pre-harvest interval (PHI) for pesticides. Therefore, it is important to update the data on the population’s real consumption value to obtain a true estimate of the risk of actual exposure to pesticides. It is impotent to continue with the pesticide residues program to reduce exposure to residues that cause long-term effects or immediate serious illness.

## Figures and Tables

**Figure 1 molecules-28-01343-f001:**
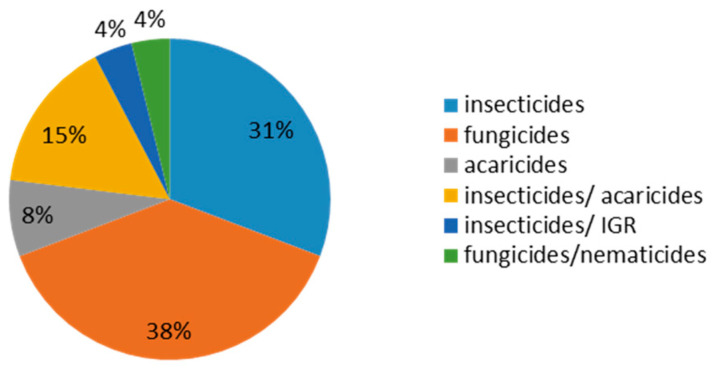
Illustration of the most important chemical groups and their percentages.

**Figure 2 molecules-28-01343-f002:**
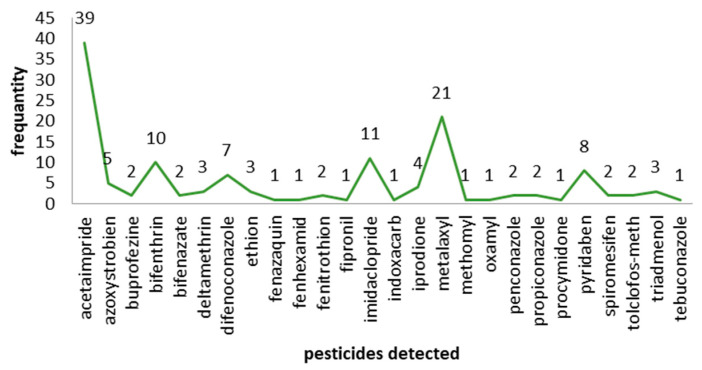
Frequency of occurrence of pesticides.

**Figure 3 molecules-28-01343-f003:**
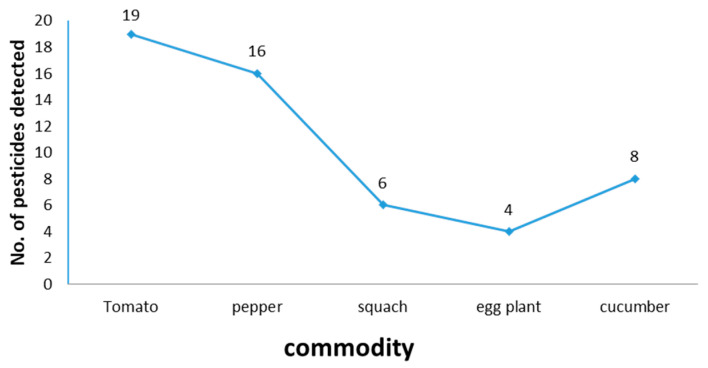
Demonstrates the contamination with pesticide residues.

**Figure 4 molecules-28-01343-f004:**
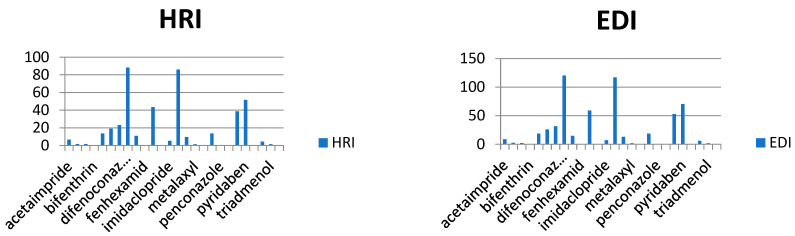
Estimation of EDI and HRI for pesticide residues detected in tomatoes.

**Figure 5 molecules-28-01343-f005:**
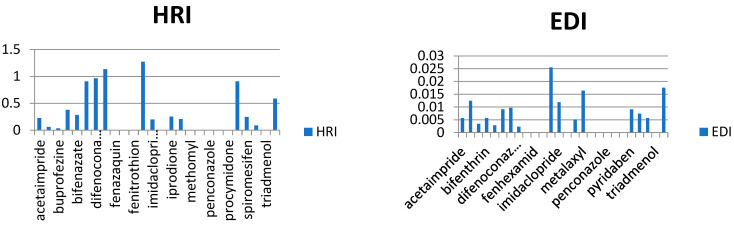
Estimation of EDI and HRI for pesticide residues detected in sweet peppers.

**Figure 6 molecules-28-01343-f006:**
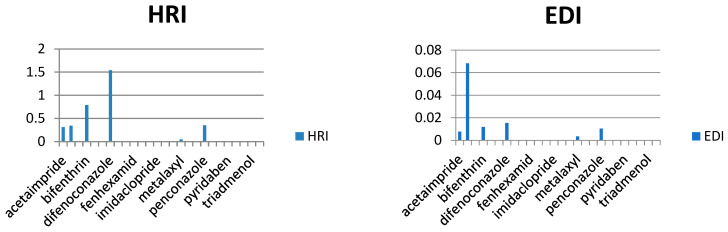
Estimation of EDI and HRI for pesticide residues detected in squash.

**Figure 7 molecules-28-01343-f007:**
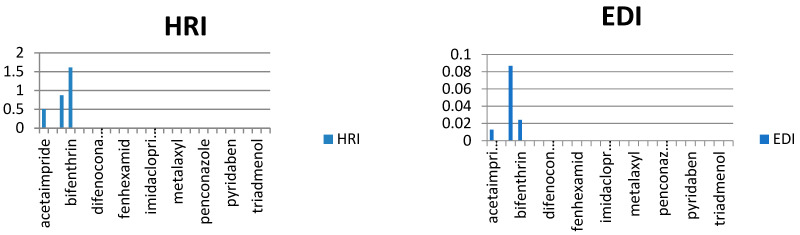
Estimation of EDI and HRI for pesticide residues detected in eggplant.

**Figure 8 molecules-28-01343-f008:**
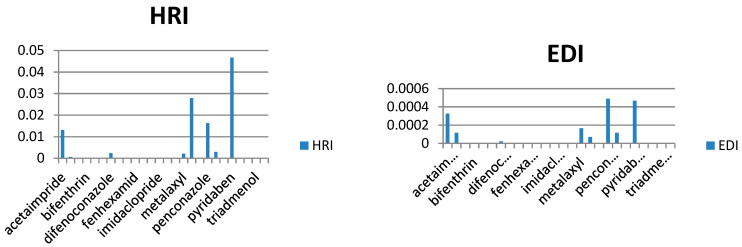
Estimation of EDI and HRI for pesticide residues detected in cucumber.

**Figure 9 molecules-28-01343-f009:**
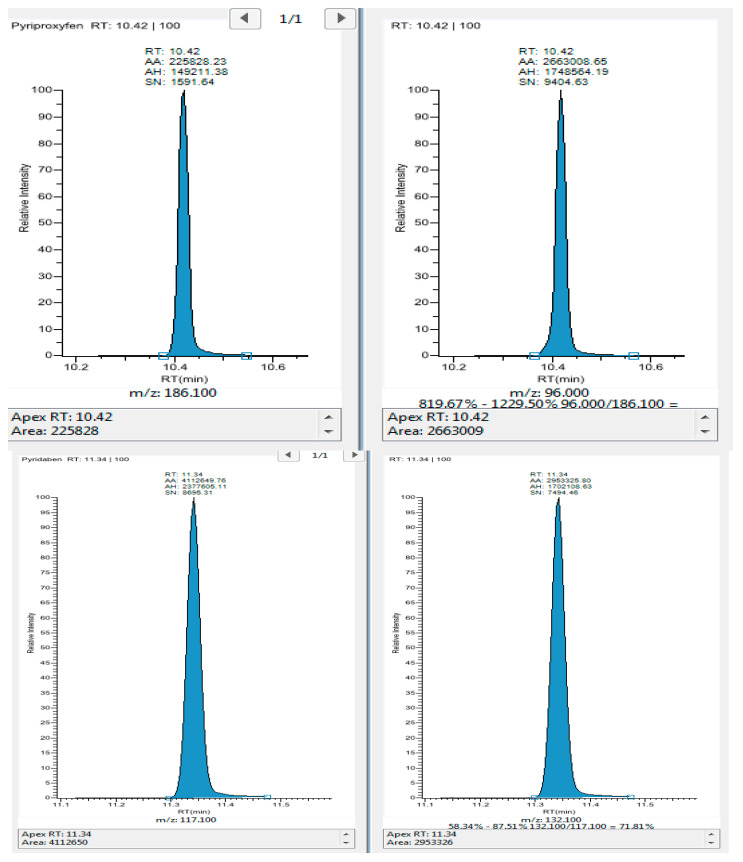
Representative multi-reaction monitoring SRM chromatograms for (1) Pyriprofexen and Pyridaben, spiked at a level of 0.100 µg/g in tomato and extracted using the sample preparation protocol reported (GC MSMS peaks).

**Table 1 molecules-28-01343-t001:** Demonstrates the frequency of occurrence of pesticides.

Pesticide	Tomato	Pepper	Squash	Eggplantt	Cucumber	Frq.
Acetaimprid	(0.017–0.347)	(0.011–0.358)	(0.011–0.118)	(0.008–0.085)	(0.018–0.209)	39
Azoxystrobien	(0.17 9–0.318)	0.216	0.39		0.054	5
Buprofezine		0.056		0.827		2
Bifenthrin	(0.03–0.362)	(0.064–0.145)	(0.01–0.125)	0.23		10
Bifenazate	0.1	0.052				2
Deltamethrin	(0.155–0.016)	0.16				3
Difenoconazole	(0.07–0.261)	(0.158–0.178)	(0.058–0.188)		0.012	7
Ethion	(0.125–0.137)	0.044				3
Fenazaquin	0.037					1
Fenhexamid	0.017					1
Fenitrothion	(0.159–0.161)					2
Fipronil		0.45				1
Imidaclopride	(0.076–0.38)	(0.018–0.721)				11
Indoxacarb	0.382					1
Iprodione	(0.178–0.0305)	0.086				4
Metalaxyl	(0.04–0.117)	(0.08–0.52)	(0.007–0.03)	0.005	(0.007–0.267)	21
Methomyl					0.026	1
Oxamyl	0.007					1
Penconazole			0.058		0.209	2
Propiconazole	(0.047–0.107)					2
Procymidone					0.053	1
Pyridaben	0.378	(0.018–0.36)			(0.08–0.328)	8
Spiromesifen		(0.06–0.196)				2
Tolclofos-meth	0.198	0.096				2
Triadmenol	(0.011–0.116)					3
Tebuconazole		0.309				1
Frq.	19	16	6	4	8	

**Table 2 molecules-28-01343-t002:** Estimated food consumption rate (g/day) in food basket: The Global Environment Monitoring System/Food Contamination Monitoring and Assessment Program (GEMS/Food).

Commodity	Consumption in the Middle East Grams per Person per Day
Tomato	81.5
Sweet Pepper	3.4
Squash	10.5
Eggplant	6.3
Cucumber	4.8

**Table 3 molecules-28-01343-t003:** Estimation of EDI and HRI for pesticide residues detected in Tomato, Sweet pepper, Squash and Cucumber samples.

Detected Pesticide	ADI	Tomato				Sweet Pepper				Squash				Eggplant				Cucumber			
		MRL	Mean	EDI	HRI	MRL	Mean	EDI	HRI	MRL	Mean	EDI	HRI	MRL	Mean	EDI	HRI	MRL	Mean	EDI	HRI
acetaimpride	0.025	0.5	0.12	0.163	6.52	0.1	0.1	0.0056	0.226	0.2	0.044	0.008	0.309	0.2	0.12	0.013	0.504	0.3	0.14	0.0003	0.013
azoxystrobien	0.2	3	0.25	0.34	1.697	3	0.22	0.0125	0.062	1	0.39	0.068	0.341	-	-	-	-	1	0.05	0.000117	0.001
buprofezine	0.1	0.3	0.12	0.163	1.63	0.5	0.06	0.0034	0.034	-	-	-	-	0.3	0.827	0.086	0.868	-	-	-	-
bifenthrin	0.015	-	-	-	-	0.5	0.1	0.0056	0.378	0.01	0.068	0.012	0.787	0.3	0.23	0.024	1.61	-	-	-	-
bifenazate	0.01	0.5	0.1	0.134	13.58	3	0.05	0.0028	0.283	-	-	-	-		-	-	-	-	-	-	-
deltamethrin	0.01	0.07	0.14	0.19	19.02	0.7	0.16	0.0091	0.907		-	-	-		-	-	-	-	-	-	-
difenoconazole	0.01	2	0.17	0.23	23.09	0.9	0.17	0.0096	0.963	0.2	0.088	0.015	1.54	-	-	-	-	0.3	0.01	2.33 × 10^−5^	0.002
ethion	0.002	0.01	0.13	0.176	88.29	0.01	0.04	0.0023	1.13	-	-	-	-		-	-	-	-	-	-	-
fenazaquin	0.005	0.5	0.04	0.054	10.86	-	-	-	-	-	-	-	-	-	-	-	-	-	-	-	-
fenhexamid	0.2	2	0.02	0.027	0.135	-	-	-	-	-	-	-	-		-	-	-	-	-	-	-
fenitrothion	0.005	0.01	0.16	0.217	43.46	-	-	-	-	-	-	-	-	-	-	-	-	-	-	-	-
fipronil	0.0002	0.005	-	-	-	-	0.45	0.025	127.5	-	-	-	-	-	-	-	-	-	-	-	-
imidaclopride	0.06	0.5	0.23	0.312	5.206	1	0.21	0.0119	0.198	-	-	-	-	-	-	-	-	-	-	-	-
indoxacarb	0.006	0.5	0.38	0.516	86.02	-	-	-	-	-	-	-	-	-	-	-	-	-	-	-	-
iprodione	0.02	5	0.14	0.19	9.508	7	0.09	0.0051	0.255	-	-	-	-		-	-	-	-	-	-	-
metalaxyl	0.08	0.3	0.08	0.108	1.358	0.5	0.29	0.0164	0.205	0.01	0.02	0.004	0.044	0.01	0.005	0.001	0.007	0.01	0.071	0.0002	0.002
methomyl	0.0025			-	-	-	-	-	-	-	-	-	-	-	-	-	-	0.04	0.03	0.0001	0.028
oxamyl	0.001	0.01	0.01	0.013	13.58	-	-	-	-		-	-	-	-	-	-	-	-	-	-	-
penconazole	0.03	-	-	-	-	-	-	-	-	0.1	0.06	0.011	0.35	-	-	0	0	0.1	0.21	0.0005	0.016
propiconazole	0.04	-	-	-	-	-	-	-	-	-	-	-	-	-	-	-	-	0	0.05	0.000117	0.003
procymidone	0.0028	0.01	0.08	0.108	38.8	-	-	-	-	-	-	-	-	-	-	-	-	-	-	-	-
pyridaben	0.01	0.15	0.38	0.516	51.61	0.5	0.16	0.009	0.906	-	-	-	-	-	-	-	-	0.15	0.2	0.0005	0.047
spiromesifen	0.03			-	-		0.13	0.007	0.246		-	-	-		-	-	-		-	-	-
tolclofos-meth	0.064		0.2	0.271	4.244		0.1	0.006	0.088		-	-	-		-	-	-		-	-	-
triadmenol	0.05		0.05	0.067	1.358			-	-		-	-	-		-	-	-		-	-	-
tebuconazole	0.03			-	-		0.31	0.018	0.585		-	-	-		-	-	-		-	-	-

**Table 4 molecules-28-01343-t004:** LC-MS/MS retention times and multi reaction monitoring MRM transitions for the LC amenable pesticides.

Pesticide	Application	Parent	CV (V)	Product 1	CE (eV)	Product 2	CE (eV)	RT
3,4,5-Trimethacarb	Insecticide	194.1	22	137.1	12	122.1	26	5.41
Acephate	Insecticide	184.1	17	143.0	8	125.1	18	1.47
Acetamiprid	Insecticide	223.0	34	126.0	20	56.1	15	3.41
Alachlor	Herbicide	271.1	28	162.1	20	238.1	11	6.30
Aldicarb	Acaricide	213.1	30	89.1	16	116.1	11	3.98
Aldicarb sulfone	Metabolite	223.0	31	148.0	10	86.0	14	2.04
Aldicarb sulfoxide	Metabolite	207.0	22	89.0	14	132.0	10	1.91
Ametryn	Herbicide	228.1	38	186.1	18	68.1	36	4.88
Anilazine	Fungicide	274.9	46	153.0	26	178.0	24	5.98
Anilofos	Herbicide	367.9	30	124.9	34	198.9	15	6.57
Atraton	Herbicide	212.0	40	170.1	18	100.0	28	3.96
Atrazine	Herbicide	216.1	39	174.1	18	96.1	23	5.20
Atrazine-desethyl	Metabolite	188.0	34	146.0	16	78.9	26	3.69
Azaconazole	Fungicide	300.0	34	159.0	28	231.1	18	5.44
Azinphos-ethyl	Insecticide	346.0	16	132.0	16	77.1	36	6.20
Azinphos-methyl	Insecticide	318.0	20	160.0	8	261.0	8	5.56
Azoxystrobin	Fungicide	404.0	28	372.0	15	329.0	30	5.73
Benalaxyl	Fungicide	326.1	26	148.0	20	91.0	34	6.62
Bendiocarb	Insecticide	224.1	26	167.0	8	109.0	18	4.62
Benfluralin	Herbicide	336.0	34	57.0	18	236.0	15	no
Benfuracarb	Insecticide	411.1	23	195.0	23	190.0	13	7.07
Benomyl	Fungicide	291.0	22	160.0	28	192.0	16	5.50
Boscalid	Fungicide	342.9	41	307.0	20	139.9	20	5.90
Buprofezin	Insecticide	306.1	31	201.0	12	57.4	20	6.96
Butachlor	Herbicide	312.2	26	57.3	22	238.2	12	7.17
Cadusafos	Insecticide	271.1	28	159.0	16	131.0	22	6.88
Carbaryl	Insecticide	202.0	28	145.0	22	117.0	28	4.86
Carbendazim	Fungicide	192.1	33	160.1	18	132.1	28	2.20
Carbofuran	Insecticide	222.1	34	165.1	16	123.0	16	4.63
Carbosulfan	Insecticide	381.0	40	118.0	22	76.0	34	7.89
Carboxin	Fungicide	236.0	34	143.0	16	87.0	22	4.79
Chlorfenvinphos	Acaricide	358.9	28	155.0	12	99.0	30	6.65
Chlorpropham	Herbicide	214.1	18	172.0	8	154.0	18	6.01
Chlorpyrifos	Insecticide	349.9	36	97.0	32	198.0	20	7.35
Chlorpyriphos-methyl	Insecticide	321.8	34	125.0	20	289.9	16	6.87
Clethodim	Herbicide	360.0	32	164.0	18	268.1	12	7.02
Coumaphos	Insecticide	363.0	32	307.0	16	289.0	24	6.60
Cyanazine	Herbicide	241.0	41	214.0	17	96.0	25	4.39
Cyanofenphos	Insecticide	304.0	34	157.0	22	276.0	12	6.57
Cymoxanil	Fungicide	199.0	23	128.0	8	111.0	18	3.58
Deltamethrin	Insecticide	505.9	28	280.9	12	93.2	46	7.64
Desmetryn	Herbicide	214.1	38	172.1	20	82.1	30	4.26
Diazinon	Insecticide	305.1	31	169.0	22	96.9	35	2.55
Dichlorvos	Acaricide	221.0	34	109.0	22	79.0	34	4.53
Dicrotophos	Insecticide	238.0	28	112.0	10	193.0	10	2.97
Diethofencarb	Fungicide	268.0	28	226.0	10	124.0	40	5.71
Difenoconazole	Fungicide	406.0	46	251.1	25	111.1	60	6.90
Dimethoate	Acaricide	230.1	24	125.0	20	199.0	10	3.32
Diniconazole	Fungicide	326.1	46	70.2	25	159.0	34	6.87
Disulfoton	Acaricide	274.9	16	89.0	20	61.1	35	6.80
Disulfoton-sulfone	Metabolite	307.1	24	97.1	28	153.1	12	5.16
Disulfoton-sulfoxide	Metabolite	291.0	24	185.0	14	97.0	31	5.08
Diuron	Herbicide	233.0	34	72.1	18	46.3	14	5.37
Epoxiconazole	Fungicide	330.0	34	121.0	22	101.0	50	6.27
Ethion	Acaricide	284.9	25	199.1	10	97.0	46	5.22
Famphur	Insecticide	326.0	32	93.0	31	217.0	20	5.19
Fenamiphos	Nematicide	304.1	36	217.1	24	202.1	36	6.39
Fenarimol	Fungicide	331.0	46	268.0	22	81.0	34	6.26
Fenazaquin	Acaricide	307.2	36	57.2	25	161.0	19	7.70
Fenhexamid	Fungicide	302.1	41	97.2	22	55.3	38	6.22
Fenitrothion	Insecticide	278.0	38	109.1	20	79.1	34	6.06
Fenobucarb	Insecticide	208.0	22	94.9	14	152.0	8	5.72
Fenoxycarb	Insecticide	302.1	28	88.0	20	116.1	11	6.45
Fenpropathrin	Insecticide	350.1	24	125.0	14	97.0	34	7.51
Fenthion	Insecticide	279.1	36	169.1	16	247.1	13	6.57
Fonofos	Insecticide	247.1	24	109.0	20	137.0	10	6.60
Heptenophos	Insecticide	251.0	26	127.0	14	125.0	14	5.43
Hexaconazole	Fungicide	314.0	40	70.1	22	159.0	28	6.74
Imazalil	Fungicide	297.0	40	159.0	22	69.0	22	5.03
Imidacloprid	Insecticide	256.1	34	175.1	20	209.1	15	3.08
Indoxacarb	Insecticide	528.0	34	150.0	22	203.0	40	6.91
Iprobenphos	Fungicide	289.0	18	91.0	20	205.0	10	6.47
Iprodione	Fungicide	330.0	21	244.7	16	288.0	15	6.40
Isocarbofos	Insecticide	291.1	21	121.1	30	231.1	13	5.39
Kresoxim-methyl	Fungicide	314.1	24	116.0	12	206.0	7	6.50
Linuron	Herbicide	249.1	31	160.1	18	181.1	16	5.75
Malathion	Acaricide	331.0	20	127.0	12	99.0	24	5.95
Metalaxyl	Fungicide	280.1	26	220.1	13	192.1	17	6.27
Metamitron	Herbicide	203.1	34	175.1	16	104.0	22	3.25
Methacrifos	Acaricide	241.1	20	125.0	20	209.1	8	5.47
Methidathion	Insecticide	303.0	18	85.1	20	145.0	10	5.45
Methiocarb	Acaricide	226.0	28	121.0	22	169.0	10	5.83
Methomyl	Insecticide	163.0	26	88.0	10	106.0	10	2.34
Metolachlor	Herbicide	284.1	26	176.1	25	252.1	15	6.33
Metolcarb	Insecticide	166.0	20	109.0	12	94.1	27	4.29
Metribuzin	Herbicide	215.0	41	131.0	18	89.0	20	4.53
Mevinphos	Acaricide	225.1	24	127.1	15	193.1	8	3.37
Monocrotophos	Acaricide	224.1	26	127.1	16	98.1	12	2.71
Myclobutanil	Fungicide	289.1	34	70.2	18	125.1	32	6.08
Omethoate	Acaricide	214.1	26	125.1	22	183.1	11	1.76
Oxadixyl	Fungicide	279.0	40	219.0	10	132.0	34	4.32
Oxamyl	Insecticide	237.0	21	72.0	10	90.0	10	2.13
Paclobutrazol	Growth Regulator	294.1	36	125.1	38	70.2	20	5.95
Penconazole	Fungicide	284.0	34	70.1	16	159.0	34	7.35
Pendimethalin	Herbicide	282.2	21	212.2	10	194.1	17	8.04
Phenmedipham	Herbicide	301.0	34	168.0	10	136.0	22	5.57
Phenthoate	Insecticide	321.0	18	163.0	12	135.0	20	6.47
Phorate	Insecticide	261.0	17	75.0	12	97.0	32	6.74
Phorate sulfone	Metabolite	293.0	24	96.9	30	115.0	24	5.20
Phosmet	Insecticide	318.0	28	160.0	22	77.0	46	4.22
Phosphamidon	Insecticide	300.1	28	174.1	14	127.1	25	4.40
Phoxim	Insecticide	299.0	22	129.0	13	153.0	7	6.69
Pirimicarb	Insecticide	239.1	34	72.0	18	182.1	15	3.55
Pirimiphos-ethyl	Insecticide	334.1	42	198.1	23	182.1	25	7.09
Probenazole	Fungicide	224.0	22	41.5	10	196.1	13	4.38
Procloraz	Fungicide	376.0	22	307.1	16	70.1	34	6.53
Procymidone	Fungicide	284.1	42	67.1	28	256.1	17	8.13
Profenofos	Insecticide	372.9	36	302.6	20	127.9	40	7.12
Promecarb	Insecticide	208.1	26	151.0	9	109.0	15	5.94
Propachlor	Herbicide	212.1	31	170.1	14	94.1	25	5.31
Propetamphos	Insecticide	282.0	17	138.0	20	156.0	12	6.07
Propham	Herbicide	180.0	14	138.0	8	120.0	16	5.15
Propiconazole	Fungicide	342.0	46	69.0	22	159.0	34	6.65
Propoxur	Insecticide	210.0	21	111.0	16	168.0	10	4.58
Pyracarbolid	Fungicide	218.1	32	125.1	18	97.1	28	4.66
Pyraclostrobin	Fungicide	388.1	31	163.0	25	193.9	12	6.70
Pyrazophos	Fungicide	374.0	44	222.1	22	194.0	32	6.75
Pyroquilon	Fungicide	174.0	41	132.0	23	117.0	30	4.49
Quinalphos	Acaricide	299.0	24	162.9	24	96.9	30	6.47
Quinmerac	Herbicide	222.2	28	204.2	15	141.1	30	3.36
Rotenone	Insecticide	395.0	46	213.1	24	192.1	24	6.39
Simazine	Herbicide	202.0	40	124.0	16	96.0	22	4.57
Simetryn	Herbicide	214.0	41	124.0	20	95.9	25	4.27
Spiromesifen	Insecticide	371.1	16	273.1	10	255.1	24	7.43
Spiroxamine	Fungicide	298.0	38	144.0	20	100.0	32	5.44
Sulfotep	Insecticide	323.0	28	97.0	32	171.0	15	6.51
Terbutryn	Herbicide	242.1	40	186.1	20	91.0	28	5.49
Thiacloprid	Insecticide	253.0	41	126.0	20	90.1	40	3.76
Thiamethoxam	Insecticide	292.0	28	211.2	12	132.0	22	2.56
Thiophanate	Fungicide	371.0	28	151.0	22	93.1	50	5.37
Tolcofos methyl	Fungicide	301.1	41	125.0	17	174.9	29	6.8
Triadimefon	Fungicide	294.1	31	69.3	20	197.2	15	5.94
Triadimenol	Fungicide	296.1	21	70.2	10	99.1	15	6.15
Triazophos	Acaricide	314.1	31	161.9	18	118.9	35	6.12
Vamidothion	Acaricide	288.0	28	146.0	10	118.0	28	3.38
Vernolat	Herbicide	204.1	28	128.1	11	86.1	14	6.83

CV = cone voltage; CE = collision energy; Rt: Retention Time.

**Table 5 molecules-28-01343-t005:** GC-MS/MS retention times and multi reaction monitoring SRM transitions for the LC amenable pesticides.

Pesticide	Application	Quantitation m/z	CE (eV)	Confirmation m/z	CE (eV)	RT (min)
Acephate	Insecticide	136.01 > 42.00	10	136.01 > 94.01	15	7.42
Alachlor	Herbicide	161.07 > 146.06	12	188.08 > 160.07	10	12.58
Atrazine	Herbicide	215.09 > 173.08	10	215.09 > 200.09	10	10.65
Azinphos-ethyl	Acaricide	132.01 > 77.01	20	160.02 > 132.01	5	19.18
Benfluralin	Herbicide	292.10 > 160.05	21	292.10 > 264.09	10	9.54
Bifenthrin	Acaricide	181.05 > 153.05	6	181.05 > 166.05	15	17.86
Boscalid	Fungicide	342.03 > 140.01	15	344.03 > 142.01	15	20.95
Bromophos-ethyl	Acaricide	358.89 > 302.91	20	358.89 > 330.90	10	14.58
Buprofezin	Chitin synthesis inhibitors	172.09 > 57.03	10	249.13 > 193.10	10	15.88
Butralin	Herbicide	266.14 > 190.10	15	266.14 > 220.11	15	13.56
Cafenstrole	Herbicide	100.04 > 72.03	15	188.08 > 119.05	15	20.21
Carbaryl	Acaricide	144.06 > 115.05	20	144.06 > 116.05	20	12.54
Chlordane	Insecticide	372.81 > 265.87	18	374.81 > 267.87	15	14.67
Chlorpropham	Herbicide	213.00 > 127.00	5	213.00 > 171.00	5	9.66
Cyfluthrin	Insecticide	163.02 > 91.01	12	163.02 > 127.02	10	20.08
Cypermethrin	Acaricide	163.03 > 127.02	10	181.03 > 152.03	25	20.66
Cyprodinil	Fungicide	224.13 > 208.12	20	225.13 > 210.12	18	14.08
Deltamethrin	Insecticide	252.99 > 93.00	18	252.99 > 173.99	18	22.19
Diazinon	Acaricide	137.05 > 84.03	10	304.10 > 179.06	15	10.09
Dimethachlor	Herbicide	197.08 > 148.06	10	199.08 > 148.06	10	12.06
Diniconazole	Fungicide	268.06 > 232.05	15	270.06 > 234.05	15	16.18
Dioxathion	Acaricide	125.00 > 97.00	15	125.00 > 141.00	15	10.78
Edifenphos	Fungicide	173.01 > 109.01	15	310.03 > 173.01	10	16.77
Ethion	Acaricide	230.99 > 202.99	15	383.99 > 230.99	10	16.18
Ethoprophos	Insecticide	158.00 > 80.90	15	158.00 > 114.00	5	9.58
Fenarimol	Fungicide	139.01 > 111.01	15	219.02 > 107.01	15	19.26
Fenobucarb	Insecticide	121.07 > 77.05	15	150.09 > 121.07	10	9.18
Fenpropathrin	Acaricide	181.09 > 152.07	23	265.13 > 210.10	15	18.06
Fipronil	Acaricide	212.97 > 177.98	16	366.95 > 212.97	25	13.94
Fluopicolide	Fungicide	208.80 > 182.00	20	261.00 > 175.00	24	16.94
Formothion	Acaricide	126.00 > 93.00	8	172.00 > 93.00	5	11.88
Imazalil	Fungicide	173.03 > 145.02	20	215.04 > 173.03	15	18.22
Iprodione	Fungicide	187.02 > 124.01	20	187.02 > 159.02	40	17.58
Isoprothiolane	Fungicide	290.06 > 118.03	15	290.06 > 204.05	15	15.28
Kresoxim-methyl	Fungicide	206.09 > 116.05	15	206.09 > 131.06	15	15.34
Lactofen	Herbicide	344.04 > 223.02	15	344.04 > 300.03	15	18.88
Malathion	Acaricide	127.01 > 99.01	10	173.02 > 127.01	10	13.05
Mecarbam	Acaricide	226.04 > 198.03	5	329.05 > 160.03	10	14.23
Mepanipyrim	Fungicide	222.11 > 207.10	15	223.11 > 208.10	15	14.26
Metalaxyl	Fungicide	249.13 > 190.10	10	249.13 > 249.13	5	12.56
Metamitron	Herbicide	202.09 > 174.07	5	202.09 > 186.08	10	10.42
Methabenzthiazuron	Herbicide	164.05 > 136.04	12	164.05 > 164.05	10	9.84
Methamidophos	Acaricide	141.00 > 95.00	10	141.00 > 126.00	5	5.77
Methidathion	Insecticide	124.98 > 98.99	22	144.98 > 84.99	10	14.65
Methiocarb	Acaricide	168.06 > 109.04	15	168.06 > 153.06	15	12.98
Metribuzin	Herbicide	198.08 > 82.03	20	198.08 > 110.05	20	12.46
Mevinphos	Acaricide	127.03 > 109.02	10	192.04 > 127.03	12	7.32
Monocrotophos	Acaricide	127.03 > 95.03	20	127.03 > 109.03	25	9.94
Omethoate	Acaricide	110.01 > 79.01	15	156.02 > 110.01	10	9.05
Penconazole	Fungicide	248.06 > 157.04	25	248.06 > 192.04	15	14.09
Pendimethalin	Herbicide	252.12 > 162.08	12	252.12 > 191.09	12	13.86
Phosalone	Acaricide	181.99 > 111.00	15	181.99 > 138.00	10	18.56
Phosphamidon	Insecticide	227.05 > 127.03	15	264.06 > 193.04	15	11.88
Pirimicarb	Insecticide	166.10 > 96.06	10	238.14 > 166.10	15	11.95
Probenfos	Insecticide	204.07 > 122.04	15	218.89 > 182.91	15	11.72
Procymidone	Fungicide	283.02 > 96.01	15	283.02 > 255.02	10	14.56
Profenofos	Insecticide	138.98 > 96.98	8	338.94 > 268.95	20	15.37
Propachlor	Herbicide	176.06 > 120.04	10	196.07 > 120.04	10	9.45
Propanil	Herbicide	217.01 > 161.00	10	219.01 > 163.00	10	12.16
Propargite	Acaricide	135.06 > 107.05	15	350.16 > 201.09	10	17.24
Propoxur	Acaricide	110.06 > 64.03	10	152.08 > 110.06	10	9.02
Pyrimethanil	Fungicide	198.11 > 158.09	30	198.11 > 183.10	15	11.28
Pyriproxyfen	Juvenile hormone mimics	226.10 > 186.10	12	136.10 > 96.00	10	10.45
Pyridaben	Acaricide	147.10 > 117.10	20	147.10 > 132.10	12	11.35
Quinalphos	Acaricide	146.03 > 118.02 1	15	157.03 > 129.02	13	14.29
Spiromesifen	Insecticide	371.24 > 273.15	15	371.24 > 255.64	25	18.42
Spiroxamine	Fungicide	100.09 > 58.05	15	100.09 > 72.06	15	12.89
Tefluthrin	Insecticide	177.02 > 127.02	20	197.03 > 141.02	15	11.27
Tetradifon	Acaricide	226.93 > 198.94	18	353.88 > 158.95	15	18.56
Tolclofos-methy	Fungicide	264.96 > 92.99	20	264.96 > 249.96	15	12.34
Triazophos	Acaricide	161.03 > 134.03	10	257.05 > 162.03	10	16.55
Trifluralin	Herbicide	264.09 > 160.05	15	306.10 > 264.09	15	9.87
Vinclozolin	Fungicide	100.09 > 58.05	15	100.09 > 72.06	15	12.35

CV = cone voltage; CE = collision energy; Rt: Retention Time.

## Data Availability

Not applicable.

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
