# Peer review of "Risk Assessment of Pesticide Residues by GC-MSMS and UPLC-MSMS in Edible Vegetables"

_molecules, 2023, doi:10.3390/molecules28031343_

Round 1
Reviewer 1 Report
The Manuscript ID Molecules - 2122994 “Risk Assessment of Pesticide Residues by GC-MSMS and UPLC-MSMS in Edible Vegetables” describes a study filling a couple of useful purposes. The use of pesticides leads to an increase in agricultural production but also causes harmful effects on human health when excessively used. For safe consumption, pesticide residues should be below the maximum residual limits (MRLs). The study showed the most common pesticides and the type of crops with the highest number of pesticide residues. The study will help in understanding of the most applied pesticides on vegetables and as the most common polluted crops in the study areas. However, the paper presents several flaws, and requires major revisions.
Line 12: The sentence is incomplete. ……a substantial increase of what? Revise the sentence.
Line 13: Add “the” between for and control; add “of” between control and pests; add “crop” between reduce and yields.
Line 13: Add “public” between major and health.
Line 15: Replace “methodology” with “method”
Lines 14 – 19: the sentence on the objective of the study is too long. Needs to be summarized,
Line 22: Frequantity is not the right term to use; replace it. It is also not clear what the values 39, 21, 11, 10, 8, 7, and 5 represent. Are these frequencies? Are these quantities? Please revise for clarity.
Lines 43 – 43: Which drug? This sentence is misplaced.
Lines 53 – 56: This paragraph is of no value to the manuscript since the methods are widely used currently for detection of pesticide residues. These are not novel techniques and also the authors were not testing the efficiency of the methods.
Lines 66 – 71: The end of the Introduction should include clear aims and numbered predictions, so to put the study in a more hypothesis-driven context.
- The numbers of the citations should be in square brackets. Revise throughout the document.
Line 97: Why are the authors starting with figure 2 instead of figure 1? Revise the numbering of figures. In addition the figure legends are not self-explanatory. Revise all the figure legends.
- The results section requires overhaul for flow of information.
Section 2.1: How many of the pesticide residues detected were above the MRL values? How many were in the “moderately hazardous (II),” and how many were in the “extremely hazardous (Ib)” class (WHO).
Section 2.1: What was the quality index of the various types of vegetables from different cities?
Lines 75 – 79: These are not results. Revise.
Section 2.2: Lines 127 – 153- these are neither results nor discussion. Lines 142 – 153 are part of the methods for the study.
Lines 157 – 160: These are not results?
Lines 174 – 184: The figure legends are not self-explanatory and require to be elaborated.
- No discussion provided in the manuscript
Line 213: Replace “Area of study” with “Study area”
Lines 221 – 223: Separate the number of samples based on the seasons. This could help to determine the effect of seasons on pesticide residues.
Lines 267 – 271: These are not methods.
Lines 317 – 322: These are not conclusions; conclusions are usually based on the objectives of the study.
Reviewer 2 Report
ADDITIONAL COMMENTS
“Risk Assessment of Pesticide Residues by GC-MSMS and UPLC-MSMS in Edible Vegetables”
In the Results section
Table 3 and 5 are not aesthetically pleasing. It is good that the tables are understandable. They are difficult to read.
No information was found in this section on the results of the validation of the analysis methods.
In the Materials and Methods section
There is no information on quantification of pesticides. The content of pesticides is strictly monitored and there are regulatory documents that the authors have cited. Since they are developing an analysis method, there is no information on the validation of the methods they use (GC-MSMS and UPLC-MSMS). This includes accuracy, precision, LOD and LOQ….
In the Compound identification section
The authors use SRM or MRM for identification. Describe in detail how the identification of the analyzed pesticides is carried out.
There are confusing data: in figure 1 you give a chromatogram of Pyriprofexen but in table 1 it is not there. In the figure, the compound is written as Pyriproxifen and the text below the figure is Pyriprofexen. Which is true?
Also, the figure is not clear:
You give an ion of 233 and the MRM is 322, which is correct? The molecular mass of the same compound is 321.369 g/mol.
The same trend was observed with Pyridaben, the chromatogram was 243 and the MRM was 365.1. Table 1 is parent ion.
The description of the figure is not clear.
Pyridaben, on the chromatogram it has tR=4.92 min, and in tablica 1 it has tR=7.71 min. What happens in this analysis? I understand from the material that the authors are developing a new method for pesticide analysis. This difference in tR indicates that you have not optimized the chromatographic separation process. When optimizing the separation process, the results must be reproducible, which is not proven with accuracy and precision.
In the Conclusions section
The information in Conclusions is not complete. Information about the analysis and what the authors achieved is missing. It is not related to the stated purpose of the study. It needs to be redone.
In the References section
The description of the references does not meet the requirements of the journal (6, 12-15). In most cases, the DOI is missing.
The manuscript needs to be revised - major revision.
Round 2
Reviewer 1 Report
Lines 94 -112: The sentences for this paragraph are too long and needs to be split. In addition, the sentences are not coherent and there is no logical flow. This section also requires extensive editing of English.
130 - 145: The sentence is too long and needs to be split. In addition, the sentence is not coherent and there is no logical flow. This sentence also requires extensive editing of English.
Figures 4 - 8: The legends for the figures are incomplete. Improve the quality of the figures.
No discussion of the results is presented in this manuscript. Please let the authors provide discussion of their results.
Reviewer 2 Report
ADDITIONAL COMMENTS
“Risk Assessment of Pesticide Residues by GC-MSMS and UPLC-MSMS in Edible Vegetables”
In the Results section
Table 5. Do you have the opportunity to make it like everyone else. It is large and has a lot of information. This makes reading the information difficult.
In the References section
The description of the references does not meet the requirements of the journal (13, 15-18). Article subject is missing.
For example:
Ref. 13
13. Pizzutti, I. R.; Kok, A.; Zanella, R.; Adaime, M. B.; Hiemstra, M.; Wickert, C.; Prestes, O. D; (2007). Method validation for the analysis of 169 pesticides in soya grain, without clean up, by liquid chromatography–tandem mass spectrometry using positive and negative electrospray ionization. Chromatogr. A, 1142, 123.
